# Immunotherapy during the Immediate Perioperative Period: A Promising Approach against Metastatic Disease

Elad Sandbank [1] , Anabel Eckerling [1] , Adam Margalit [2], Liat Sorski [1] and Shamgar Ben-Eliyahu [1,2,*]

1    Neuro-Immunology Research Unit, School of Psychological Sciences, Tel Aviv University,
     Tel Aviv 69978, Israel; eladsandbank@mail.tau.ac.il (E.S.); anabele@mail.tau.ac.il (A.E.);
     liatsors@tauex.tau.ac.il (L.S.)
2    Sagol School of Neuroscience, Tel Aviv University, Tel Aviv 69978, Israel; adammargalit@mail.tau.ac.il
*    Correspondence: shamgar@tauex.tau.ac.il; Tel.: +972-36407266

**Abstract:** Tumor excision is a necessary life-saving procedure in most solid cancers. However, surgery and the days before and following it, known as the immediate perioperative period (IPP), entail numerous prometastatic processes, including the suppression of antimetastatic immunity and direct stimulation of minimal residual disease (MRD). Thus, the IPP is pivotal in determining long-term cancer outcomes, presenting a short window of opportunity to circumvent perioperative risk factors by employing several therapeutic approaches, including immunotherapy. Nevertheless, immunotherapy is rarely examined or implemented during this short timeframe, due to both established and hypothetical contraindications to surgery. Herein, we analyze how various aspects of the IPP promote immunosuppression and progression of MRD, and how potential IPP application of immunotherapy may interact with these deleterious processes. We discuss the feasibility and safety of different immunotherapies during the IPP with a focus on the latest approaches of immune checkpoint inhibition. Last, we address the few past and ongoing clinical trials that exploit the IPP timeframe for anticancer immunotherapy. Accordingly, we suggest that several specific immunotherapies can be safely and successfully applied during the IPP, alone or with supporting interventions, which may improve patients' resistance to MRD and overall survival.

**Keywords:** immediate perioperative period; immunotherapy; surgery; immune checkpoint inhibitors

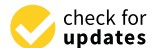



## 1. Introduction to the Immediate Perioperative Period (IPP)

Surgery is a key intervention for the treatment of solid tumors. However, whereas surgery is highly effective in achieving a complete resection of the primary tumor, a significant portion of patients bear undetected minimal residual disease (MRD) in the form of micrometastases and/or circulating malignant cells [1]. Operated cancer patients are commonly subjected to stress and inflammatory responses throughout their disease course, with peaks starting at cancer diagnosis, followed by awaiting surgery, numerous medical procedures including the surgical procedure itself, anesthesia and analgesia, hypothermia, blood loss and transfusion, and their ensuing rehabilitation and recovery [2–4]. Importantly, the biological responses to surgery and its related psychological stress were shown to promote numerous signaling cascades, including sympathetic, steroid, inflammatory, and proangiogenic responses, which in turn could lead to immune suppression and/or the direct stimulation of MRD and its microenvironment [2–4]. These responses were also shown to facilitate the progression of MRD and its ability to overcome immune surveillance and malignant cell dormancy, potentially leading to an eruption of life-threatening metastatic disease [2–4]. Thus, the immediate perioperative period (IPP), ranging from three weeks before to three weeks after tumor excision, is now believed to exert a profound impact on the risk of metastatic disease and long-term survival outcomes [2–4]. The attenuation of such deleterious processes during the IPP, through the blockade of promalignant growth

signals or the enhancement of antimetastatic immunity, may bear a significant clinical advantage for cancer patients [3–5].

Unfortunately, interventions to prevent postoperative metastatic disease, including immunotherapy, are rarely implemented during the IPP due to the confirmed and sometimes hypothesized deleterious effects of immunotherapy on patients' safety and success rates of surgery. Most tested or commonly used immunotherapeutic approaches are non-IPP, ending at least four weeks before surgery and/or initiated a minimum of four weeks following surgery. In this review, we discuss various aspects of the IPP that promote immunosuppression, and the progression of residual malignant tissue, and how these deleterious processes may also jeopardize specific aspects of perioperative immunotherapy. We discuss the safety and potential efficacy of different immunotherapies in the perioperative context and discuss which types of available immunotherapies are most suitable to use during the IPP. Last, we address the limited number of past and current clinical trials and clinical routines that exploit the IPP for anticancer immunotherapy.

## 2. Direct Effects of Perioperative Stress and/or Inflammatory Responses on Malignant Tissue

Perioperative events and related psychological states, including patients' anxiety, surgery-induced tissue damage, exposure to anesthetic and analgesic agents, blood transfusions, hypothermia, nociception, and pain, have all been shown to promote stress-inflammatory responses, mostly through the local and systemic release of catecholamines (CAs, i.e., epinephrine and norepinephrine) and prostaglandins (PGs), especially prostaglandin E2 [3,4]. While such stress-inflammatory responses can be adaptive in the context of physical challenges, such as injuries and the subsequent wound healing, the same responses during the IPP can promote primary tumor prometastatic characteristics [3,4] as well as the progression of MRD. Specifically, β-adrenergic and prostaglandin signaling were shown to promote tumor cells' epithelial-to-mesenchymal transition, proliferation, resistance to apoptosis, evasion from antitumor immune response, angiogenesis, tumor-supporting inflammation, and tumor cells' invasive and metastasis capacity [2,6]. For example, β-adrenergic signaling was shown to induce the secretion of matrix metalloproteinases and proangiogenic factors (VEGF, interleukin (IL)-6, and IL-8) in human melanoma [7,8], breast [9], and ovarian [10] cancer cells, and induce tumor epithelial-to-mesenchymal transition [11], all of which are related to an elevated metastatic risk and worse prognosis [3,12].

## 3. Synergistic Perioperative Deleterious Effects of Stress and Inflammation and Their Blockade

Notably, there is synergism between β-adrenergic and prostaglandin signaling. For example, preclinical studies report that behavioral-stress-induced β-adrenergic signaling can promote cyclooxygenase 2 (COX2) expression and prostaglandin secretion in both tumor cells and tumor-associated macrophages [13–15]. Additionally, both tumor and immune cells express adrenergic and prostaglandin receptors, and the activation of each receptor system converges on the same intracellular pathways (e.g., cAMP). Thus, during simultaneous stress and inflammatory responses, only their combined blockade can overcome the deleterious impacts of such responses. Indeed, in our translational studies, the combined use of a β-adrenergic receptor antagonist, propranolol, and a prostaglandin synthesis inhibitor, etodolac, during the IPP, was more effective than each drug alone and often the only effective intervention to improve the resistance to postoperative metastasis [16–18].

Nevertheless, the sole attenuation of stress responses before surgery, without a direct blockade of inflammatory responses, was shown to be sufficient to improve primary breast tumor prometastatic characteristics. Hiller et al. studied the use of the β-adrenergic receptor antagonist, propranolol, starting a week before surgery [19], and we employed a stress-reducing psychological intervention starting 2–3 weeks before surgery [20]. Both studies assessed primary tumor transcriptomic profiles and transcription factor activity levels in the excised breast tumors, and both found a significant improvement in primary tumor metastatic characteristics. In the psychological intervention, both adrenergic and corticoid

signaling were reduced, as well as inflammatory (NFkB) and prometastatic molecular biomarkers (e.g., GATA). In general, however, unlike the primary tumor, MRD is also subjected to postoperative inflammatory signaling, which is triggered more potently by the surgery itself. Thus, such interventions (that do not address surgery-induced inflammatory signaling) may not be sufficient to improve long-term cancer outcomes.

## 4. Immunosuppression during and following the IPP

Immunosuppression during and following the IPP is a common and established phenomenon [21–23]. Operated cancer patients are subjected to stress and anxiety, surgery and its related medical procedures (e.g., tissue damage, anesthesia, and hypothermia), and the postoperative resolution of inflammatory processes, which include systemic anti-inflammatory responses [5]. These processes have all been shown to induce immunosuppression, which can consequently increase postoperative complication rates and enable MRD, in the form of disseminated tumor cells or dormant micrometastases, to grow into overt disease [3–5,23]. An excess perioperative release of CAs and PGs characterizes most of these deleterious processes, and has been shown to directly cause the suppression of cell-mediated immunity. Specifically, in preclinical studies, CAs and PGs were shown to (i) shift the TH1/TH2 balance toward TH2 dominance [24], (ii) downregulate the number and activity of CD8+ and CD4+ effector T cells [25], (iii) upregulate suppressive regulatory T cells and myeloid-derived suppressor cells [26–28], (iv) reduce natural killer (NK) cell cytotoxicity [29–32], and (v) promote M2 macrophage polarization [15,33–35], all of which were shown to promote metastasis. Importantly, while the choice of anesthetic and analgesic agents during surgery was clearly shown to impact postoperative immunosuppression and recurrence risks in preclinical studies [3,4,36], very few randomized controlled trials have been conducted. So far, these studies have failed to demonstrate effects on long-term clinical outcomes [37], and the relevance of surgical anesthetic and analgesic strategies to the survival of cancer patients is mostly based on retrospective associative studies [38]. Nevertheless, there are more robust and causative clinical data on the immune-suppressive effects of the surgical setting [23], including anesthesia and analgesic agents [36,39], hypothermia [40,41], tissue damage, and blood loss and transfusion [42].

Perioperative immune suppression in cancer patients can be evident not only following surgery, but even beforehand [11,43]. Cancer patients experience distress and anxiety as they await surgery [4,44], which in turn can cause the suppression of antitumor immunity [45]. For example, in patients with breast or ovarian cancer, plasma cortisol levels and/or inflammatory indices (IL-6 and CRP) were elevated prior to surgery [11,46]. Non-cancer-operated patients also exhibit immune perturbations starting before surgery, including reduced leukocyte cell surface expression of MHC class II and lower counts of granulocytes and CD$^{8bri}$CD4$^{dim}$ lymphocytes [47]. Immune suppression is also observed immediately after surgery, with decreased levels of interferon (IFN) gamma and IL-12-induced production (but not plasma levels) reported in breast cancer (BC) patients the morning after surgery [11]. In lung cancer patients, elevated expression of the inhibitory PD-1/PD-L1 on immune cells (CD4+ and CD8+ T cells and NK cells) has been reported following surgery, with lower counts of T lymphocytes and NK cells [48]. After surgery, immunosuppression persists for days and weeks, with decreased NK cell numbers and/or activity lasting up to 1–2 months following surgery, as reported in colorectal cancer patients [49,50]. Overall, such perioperative immune perturbations are reported postoperatively in patients exhibiting several types of solid malignancies, and predict long-term cancer outcomes [51]. Circumventing these deleterious immune perturbations by potentially harnessing existing or novel immune-modulating approaches, during the critical IPP, may be advantageous over their application after the facilitation of immunosuppression [5].

## 5. Immunotherapy—Overview

Immunotherapy has seen highs and lows over the years and has recently regained momentum via novel regimens of immune checkpoint inhibitors. Immunotherapy refers

to a range of therapeutic interventions that enhance or preserve the immune system's ability to recognize and abolish cancer cells. Unlike traditional cancer treatments, such as surgery, chemotherapy, or radiation therapy, immunotherapy aims to promote the capacity of the host to eliminate or control the disease, and thus is affected by various processes that regulate immunity. Immunotherapy achieves its goals by performing the following: (i) employing immunostimulatory agents, like cytokines [52] or Toll-like receptor agonists [53]; (ii) preserving anticancer immune functions through the blockade of immunosuppressive mediators (e.g., β-blockers or COX2 inhibitors) [22] or through the inhibition of immune checkpoint signaling (e.g., cytotoxic T lymphocyte protein 4 (CTLA-4) or programmed cell death 1 (PD-1) inhibition) [54]; (iii) directing oncolytic viruses [55] or monoclonal antibodies [56] at the malignant tissue to enhance immune responses to tumor cells, induce immunogenic cell death, and/or to attenuate prometastatic processes (e.g., a blockade of the VEGF/VEGFR-2 signaling pathway [57,58]); and (iv) other approaches which will not be discussed in this review.

### 5.1. Immunotherapy during the IPP: Feasibility and Effectiveness

In general, very few clinical studies have evaluated or are currently evaluating the safety and efficacy of immunotherapeutic agents during the IPP (see Table 1 and Figure 1) [59–63], whereas the assessment of non-IPP administration of these agents is becoming acknowledged and established as an effective approach in numerous types of cancer [64]. For example, we searched for the keywords: "cancer" and "perioperative immunotherapy" on clinicaltrials.gov (starting from 1 January 2018) and found 57 interventional studies (Table 1) in which almost half of the trials did not describe the specific timing of the treatment (48%); out of the trials which described the specific timing of treatment, the majority were non-IPP therapies (73%), and the great majority assessed immune checkpoint blockade (93%). The majority of the published literature on immunotherapy, and specifically perioperative immunotherapy in cancer patients, is currently focused on immune checkpoint inhibitors (ICIs). The rapid evolution and expansion of clinical trials assessing numerous regimens of ICIs, as monotherapy or in conjunction with other treatments, at different time points relative to surgery, and in various cancer types and cancer patient populations, position ICIs as a suitable approach for discussing the various considerations for the use of immunotherapy during the IPP. Accordingly, the following sections discuss several types of immunotherapies that are currently being tested during the IPP, and address ICIs in more detail.

### 5.1.1. Anti-Stress-Inflammatory Approach—The Inhibition of β-Adrenergic and COX2 Signaling

As others and us have indicated in preclinical studies, excess β-adrenergic signaling and prostaglandins synthesis were each shown to disrupt antimetastatic immune surveillance and cell-mediated immunity, promote metastasis, and worsen survival outcomes [4,65]. In the past two decades, we have established a novel approach that targets these deleterious responses during the IPP, employing the β-adrenergic receptor antagonist, propranolol, and the prostaglandin synthesis inhibitor, etodolac. The combined treatment was found safe during the IPP [66,67] and effective in attenuating immunosuppression and metastatic disease. Specifically, we assessed the safety of these drugs, alone and in combination, on wound healing, anastomosis strength, and abdominal muscle tensile strength, and reported no deleterious effect [66,67]. Additionally, the combined propranolol and etodolac treatment was employed by us during the IPP in several tumor models, showing enhanced antimetastatic immunity and decreased metastatic burden and/or improved survival rates in all models [17,18,32,68]. Importantly, the simultaneous administration of propranolol and etodolac was more effective than each drug alone, and often the only effective intervention to improve resistance to postoperative metastasis [16–18].

**Table 1.** Interventional clinical trials assessing perioperative immunotherapy in cancer patients according to https://clinicaltrials.gov/ (accessed on 14 April 2023).

| | Interventions | Period | Specific Timing | Title | Status | Conditions | Phase | Trial Identifier |
|---|---|---|---|---|---|---|---|---|
| 1 | Combination product: MVA-BN-Brachyury and Atezolizumab | IPP | Three weeks before and after surgery (on average) | Perioperative With MVA-BN-Brachyury and PROSTVAC For Intermediate-Risk And High-Risk Localized Prostate Cancer | Withdrawn | Prostate adenocarcinoma | 2 | NCT04020094 |
| 2 | Drug: QBECO Drug: Placebo | IPP | Days before surgery and 6 weeks after | A Study of QBECO Versus Placebo in the Treatment of Colorectal Cancer That Has Spread to the Liver | Not yet recruiting | Colorectal cancer Liver metastases | 2 | NCT05677113 |
| 3 | Drug: M7824; drug: M9241; radiation: SBRT | IPP | 0–2 weeks before surgery | Immune Checkpoint Inhibitor M7824 and the Immunocytokine M9241 in Combination With Stereotactic Body Radiation Therapy (SBRT) in Adults With Advanced Pancreas Cancer | Terminated | Pancreatic cancer Pancreatic neoplasms Metastatic pancreatic cancer | 1 \| 2 | NCT04327986 |
| 4 | Drug: Ipilimumab Drug: Nivolumab Procedure: core biopsy/cryoablation Procedure: breast surgery | IPP | 1–2 weeks before surgery and 2 weeks after | Peri-Operative Ipilimumab + Nivolumab and Cryoablation in Women With Triple-negative Breast Cancer | Recruiting | Breast cancer | 2 | NCT03546686 |
| 5 | Drug: Histamine Dihydrochloride (HDC) Drug: Interleukin-2 (IL-2) | IPP | Two weeks daily up to surgery and continuing three days after for a week | Histamine Dihydrochloride and Interleukin-2 in Primary Resectable Pancreatic Cancer | Not yet recruiting | Pancreatic cancer metastasis | 2 | NCT05810792 |
| 6 | Drug: Sintilimab injection Drug: TACE Radiation: radiotherapy | IPP | Up to 3 weeks before surgery and 4 weeks after | Perioperative Therapy for Hepatocellular Carcinoma | Terminated | Hepatocellular carcinoma | 2 | NCT04653389 |
| 7 | Drug: Camrelizumab Drug: Apatinib Mesylate Procedure: postoperative TACE Procedure: radical surgery Procedure: preoperative TACE | IPP | 1 week before surgery at most and 4–8 weeks after | Camrelizumab Combined With Apatinib Mesylate and TACE in the Perioperative Treatment of Hepatocellular Carcinoma | Recruiting | Hepatocellular carcinoma | 3 | NCT05613478 |

**Table 1.** *Cont.*

| | Interventions | Period | Specific Timing | Title | Status | Conditions | Phase | Trial Identifier |
|---|---|---|---|---|---|---|---|---|
| 8 | Drug: Camrelizumab Drug: Apatinib Mesylate Procedure: TACE treatment Procedure: radical surgery | IPP | 2–4 weeks before surgery and 4 weeks after surgery | Camrelizumab Combined With Apatinib Mesylate for Perioperative Treatment of Resectable Hepatocellular Carcinoma | Recruiting | Hepatocellular carcinoma Molecular-targeted therapy | N/A | NCT04521153 |
| 9 | Drug: Nivolumab Drug: Ipilimumab | N/S | | Immunotherapy in Patients With Early dMMR Rectal Cancer | Not yet recruiting | Cancer of the rectum | 2 | NCT05732389 |
| 10 | Drug: Nivolumab Drug: Relatlimab | N/S | | Feasibility and Efficacy of Perioperative Nivolumab With or Without Relatlimab for Patients With Potentially Resectable Hepatocellular Carcinoma (HCC) | Recruiting | Hepatocellular carcinoma | 1 | NCT04658147 |
| 11 | Drug: nab-paclitaxel AUC = 2 and carboplatin 80 mg/m$^2$ Drug: camrelizumab 200 mg Procedure: radiotherapy | Non-IPP | | Camrelizumab Combined With Neoadjuvant Concurrent Chemoradiotherapy for Resectable Locally Advanced ESCC | Not yet recruiting | Esophageal squamous cell carcinoma | 2 | NCT05650216 |
| 12 | Drug: Penpulimab Drug: Anlotinib Hydrochloride Drug: Cadonilimab Drug: chemotherapy | Non-IPP | 6–9 weeks before surgery | An Exploratory Study of Immunotherapy Combined With Anlotinib and Chemotherapy in Perioperative Treatment of LAGC | Not yet recruiting | Gastric cancer | N/A | NCT05800080 |
| 13 | Drug: Pembrolizumab | Non-IPP | Six weeks prior | Immunotherapy in MSI/dMMR Tumors in Perioperative Setting. | Recruiting | MSI/dMMR or EBV-positive gastric cancers | 2 | NCT04795661 |
| 14 | Drug: Recombinant intravesical BCG (Bacillus Calmette-G$^{u©}$rin VPM1002BC); drug: Atezolizumab; drug: Cisplatin; drug: Gemcitabine | Non-IPP | 7–11 weeks before surgery and 4 weeks after | Intravesical Recombinant BCG Followed by Perioperative Chemo-immunotherapy for Patients With MIBC | Suspended | Bladder Cancer | 2 | NCT04630730 |

| | Interventions | Period | Specific Timing | Title | Status | Conditions | Phase | Trial Identifier |
|---|---|---|---|---|---|---|---|---|
| 15 | Drug: sintilimab Drug: anlotinib | Non-IPP | At least 4–6 weeks before and 3 weeks after surgery | Sintilimab Combined With Anlotinib for Perioperative Non-small Cell Lung Cancer Based on MRD Evaluation | Recruiting | Non-small-Cell Lung Cancer | 2 | NCT05460195 |
| 16 | Drug: XELOX or SOX Drug: JS001 + XELOX or SOX | Non-IPP | Five weeks before surgery and five weeks after | Neoadjuvant Immunotherapy and Chemotherapy for Locally Advanced Esophagogastric Junction and Gastric Cancer Trial | Recruiting | Gastric cancer; stomach neoplasm | 2 | NCT04744649 |
| 17 | Drug: Toripalimab | Non-IPP | 6–8 weeks before surgery and 4–6 weeks after | Neoadjuvant Chemoradiotherapy Combined With Perioperative Toripalimab in Locally Advanced Esophageal Cancer | Recruiting | Advanced esophageal squamous cell Cancer | 2 | NCT04437212 |
| 18 | Drug: Camrelizumab; drug: Oxaliplatin; drug: S1 | Non-IPP | 3–9 weeks before surgery | Efficacy and Safety of Perioperative Chemotherapy Plus PD-1 Antibody in Gastric Cancer | Unknown status | Gastric cancer | 2 | NCT04367025 |
| 19 | Drug: Nivolumab and Ipilimumab; other: chemotherapy | Non-IPP | | Postoperative Immunotherapy vs Standard Chemotherapy for Gastric Cancer With High Risk for Recurrence (VESTIGE) | Active, not recruiting | Gastric and esophagogastric junction adenocarcinoma | 2 | NCT03443856 |
| 20 | Drug: nivolumab 4.5 mg/kg + Paclitaxel (albumin-bound-type) 260 mg/m2+ Carboplatin AUC5 | Non-IPP | Seven weeks before surgery (extrapolated from outcome measures) | Clinical Study of Neoadjuvant Chemotherapy and Immunotherapy Combined With Probiotics in Patients With Potential/Resectable NSCLC | Active, not recruiting | Non-small-cell lung cancer stage III | 1 | NCT04699721 |
| 21 | Drug: teripalimab plus chemotherapy; drug: chemotherapy plus teripalimab | Non-IPP | 8–10 weeks before surgery | Teripalimab Plus Chemotherapy in Local Advanced Esophageal Cancer | Unknown status | Squamous cell carcinoma esophageal cancer | 2 | NCT03985670 |
| 22 | Drug: Atezolizumab Procedure: conventional surgery Drug: Fluorouracil Drug: Oxaliplatin | Non-IPP | 6–8 weeks before surgery and 6 weeks after | Atezolizumab, Oxaliplatin, and Fluorouracil in Treating Patients With Esophageal or Gastroesophageal Cancer | Recruiting | Esophageal; gastroesophageal junction; adenocarcinoma AJCC | 2 | NCT03784326 |

**Table 1.** *Cont.*

| | Interventions | Period | Specific Timing | Title | Status | Conditions | Phase | Trial Identifier |
|---|---|---|---|---|---|---|---|---|
| 23 | Drug: Lenvatinib; drug: Lenvatinib Mesylate; drug: Pembrolizumab | Non-IPP | At least 4 weeks before surgery | Perioperative Lenvatinib With Pembrolizumab in Patients With Locally Advanced Nonmetastatic Clear Cell Renal Cell Carcinoma | Recruiting | Kidney cancer Renal cell cancer | 2 | NCT04393350 |
| 24 | Drug: Tislelizumab; drug: pemetrexed; drug: cis-platinum; or drug: carboplatin | Non-IPP | 7–9 weeks before surgery | A Single-arm Exploratory Study of Neoadjuvant Therapy | Recruiting | Non-small-cell lung cancer | N/A | NCT05527808 |
| 25 | Drug: Durvalumab; drug: Tremelimumab; drug: Cisplatin-based neoadjuvant chemotherapy | Non-IPP | Twelve weeks before surgery | Durvalumab (MEDI4736) and TREmelimumab in NEOadjuvant Bladder Cancer Patients (DUTRENEO) | Completed | Bladder cancer | 2 | NCT03472274 |
| 26 | Drug: Axitinib (VEGF-TKI); procedure: cytoreductive nephrectomy (CN); procedure: metastasectomy (MET); biological: Pembrolizumab | Non-IPP | 5–6 weeks before surgery and 3–6 weeks after | Pembrolizumab With or Without Axitinib for Treatment of Locally Advanced or Metastatic Clear Cell Kidney Cancer in Patients Undergoing Surgery | Recruiting | Metastatic/recurrent clear-cell renal cell carcinoma; renal cell cancer | 2 | NCT04370509 |
| 27 | Drug: Nivolumab 10 MG/ML; drug: Ipilimumab 200 MG in a 40 ML injection | Non-IPP | Seven weeks before surgery and four weeks after | Peri-operative Association of Immunotherapy (Pre-operative Association of Nivolumab and Ipilimumab, Post-operative Nivolumab Alone) in Localized Microsatellite Instability (MSI) and/or Deficient Mismatch Repair (dMMR) Oeso-gastric Adenocarcinoma | Recruiting | Localized esogastric adenocarcinoma; MSI and or dMMR | 2 | NCT04006262 |
| 28 | Drug: Nivolumab; drug: carboplatin; drug: nab-paclitaxel | Non-IPP | Eight weeks before and two weeks after surgery | A Two-arm (Phase 2) Exploratory Study of Nivolumab Monotherapy or in Combination With Nab-paclitaxel and Carboplatin in Early Stage NSCLC in China | Recruiting | Non Small Cell Lung Cancer | 2 | NCT04015778 |

**Table 1.** *Cont.*

| | Interventions | Period | Specific Timing | Title | Status | Conditions | Phase | Trial Identifier |
|---|---|---|---|---|---|---|---|---|
| 29 | Drug: neoadjuvant radiation plus SOX and PD-1 antibody | Non-IPP | 5 weeks before surgery and 3–8 weeks after | Neoadjuvant Immunotherapy and Chemoradiotherapy for Locally Advanced Esophagogastric Junction Adenocarcinoma | Not yet recruiting | Adenocarcinoma of esophagogastric junction | 2 | NCT05505461 |
| 30 | Drug: Tirelizumab; drug: Paclitaxel; drug: Carboplatin; radiation: neoadjuvant radiotherapy | Non-IPP | 7–9 weeks before surgery | Selected Chemotherapy Combined Immunotherapy Treated High Risk Patient After NCRT in Resected Locally Advanced ESCC | Recruiting | Esophageal squamous cell carcinoma | 2 | NCT05189730 |
| 31 | Drug: Toripalimab; radiation: stereotactic body radiation therapy (SBRT) | Non-IPP | 6–9 weeks before surgery (after unknown) | Neoadjuvant Radiotherapy Combined With Toripalimab for Locally Advanced Head and Neck Squamous Cell Carcinoma | Not yet recruiting | Locally advanced head and neck squamous cell carcinoma | 2 | NCT05861557 |
| 32 | Drug: camrelizumab; drug: Paclitaxel for injection (albumin-bound); drug: Cisplatin | Non-IPP | Nine weeks before surgery | Efficacy of Neoadjuvant PD-1 Blockade Plus Chemotherapy for Esophageal Squamous Cell Carcinoma | Completed | Esophageal squamous cell Carcinoma | 2 | NCT04225364 |
| 33 | Drug: Carillizumab; procedure: esophagectomy; other: samples | Non-IPP | 4–6 weeks before surgery | Carrelizumab, Chemotherapy and Apatinib in the Neoadjuvant Treatment of Resectable Esophageal Squamous Cell Carcinoma | Active, not recruiting | Esophageal squamous cell carcinoma | 2 | NCT04666090 |
| 34 | Drug: Nivolumab; drug: relatlimab; drug: Oxaliplatin; drug: Docetaxel; drug: 5-Fluorouracil (5-FU); drug: folic acid (FA) | Not described | | Perioperative Immunotherapy vs. Chemo-immunotherapy in Patients With Advanced GC and AEG | Active, not recruiting | Gastric cancer esophagogastric junction adenocarcinoma | 2 | NCT04062656 |
| 35 | Combination product: Toripalimab combined with cisplatin and paclitaxel; combination product: placebo combined with cisplatin and paclitaxel | Not described | | Perioperative Toripalimab (JS001) Combined With Neoadjuvant Chemotherapy in Patients With Resectable Locally Advanced Thoracic Esophageal Squamous Cell Carcinoma | Recruiting | Resectable locally advanced thoracic esophageal squamous cell carcinoma | 3 | NCT04848753 |

**Table 1.** *Cont.*

| | Interventions | Period | Specific Timing | Title | Status | Conditions | Phase | Trial Identifier |
|---|---|---|---|---|---|---|---|---|
| 36 | Drug: neoadjuvant immunotherapy; drug: neoadjuvant chemotherapy | Not described | | Effect of Neoadjuvant Anti-PD-1 Immunotherapy on Perioperative Analgesia and Postoperative Delirium | Recruiting | NSCLC | N/A | NCT05273827 |
| 37 | Drug: Atezolizumab; drug: Capecitabine; drug: Docetaxel; drug: Fluorouracil; drug: Leucovorin Calcium; drug: Oxaliplatin; procedure: positron emission tomography; procedure: surgical procedure | Not described | | Testing Immunotherapy (Atezolizumab) With or Without Chemotherapy in Locoregional MSI-H/dMMR Gastric and Gastroesophageal Junction (GEJ) Cancer | Not yet recruiting | Stage I-II-III-IVA—localized gastric cancer and gastroesophageal junction adenocarcinoma AJCC v8 | 2 | NCT05836584 |
| 38 | Procedure: surgery | Not described | | Safety and Feasibility of Surgery After Conversion Therapy for Locally Advanced and Advanced NSCLC | Recruiting | Pulmonary Neoplasm \| Advanced Cancer \| Locally Advanced Cancer | N/A | NCT04945928 |
| 39 | Drug: Oxaliplatin; drug: Tegafur–Gimeracil–Oteracil; drug: Sintilimab; radiation: concurrent chemoradiation; procedure: D2/R0 gastrectomy | Not described | | Perioperative Chemoimmunotherapy With/Without Preoperative Chemoradiation for Locally Advanced Gastric Cancer | Recruiting | Stomach neoplasms; esophagogastric junction disorder | 2 | NCT05161572 |
| 40 | Drug: Oxaliplatin; drug: SOX neoadjuvant; drug: Sintilimab neoadjuvant; procedure: gastrectomy plus D2 lymph node dissection; drug: SOX adjuvant; drug: Sintilimab adjuvant | Not described | | Transarterial Neoadjuvant Chemotherapy vs. Traditional Intravenous Chemotherapy For Locally Advanced Gastric Cancer With SOX + PD-1 | Recruiting | Locally advanced gastric carcinoma | 3 | NCT05593458 |

| | Interventions | Period | Specific Timing | Title | Status | Conditions | Phase | Trial Identifier |
|---|---|---|---|---|---|---|---|---|
| 41 | Drug: Toripalimab; drug: Docetaxel; drug: Fluorouracil; drug: Leucovorin; drug: Oxaliplatin | Not described | | Toripalimab Combined With FLOT —Neoadjuvant Chemotherapy in Patients With Resectable Gastric Cancer | Recruiting | Gastric cancer | Phase 2 | NCT04354662 |
| 42 | Drug: Anti-PD-1 Monoclonal Antibody JS001; drug: chemotherapy | Not described | | The Efficacy of JS001 Combined With Chemotherapy in Patients With Locally Advanced Colon Cancer | Recruiting | Colonic neoplasms | 1\|2 | NCT03985891 |
| 43 | Procedure: Hepatectomy Drug: Pembrolizumab Drug: Vactosertib | Not described | | Preoperative Immunotherapy (Pembrolizumab) for Patients With Colorectal Cancer and Resectable Hepatic Metastases | Recruiting | Metastatic malignant neoplasm in the liver; stage IV colorectal cancer | 2 | NCT03844750 |
| 44 | Drug: Serplilumab; drug: Capecitabine; drug: Oxaliplatin; drug: Celecoxib | Not described | | Serplulimab Combined With CAPEOX + Celecoxib as Neoadjuvant Treatment for Locally Advanced Rectal Cancer | Recruiting | pMMR\|MSS\|MSI-L\|locally advanced rectal carcinoma | 2 | NCT05731726 |
| 45 | Drug: Nivolumab; drug: Ipilimumab | Not described | | A Study of Immunotherapy Drugs Nivolumab and Ipilimumab in Patients w/Resectable Malignant Peritoneal Mesothelioma | Active, not recruiting | Mesothelioma\|peritoneal mesothelioma | 2 | NCT05041062 |
| 46 | Drug: Tislelizumab; drug: gemcitabine and cisplatin; radiation: modified hypofractionation | Not described | | Risk-stratification Based Bladder-sparing Modalities for Muscle-invasive Bladder Cancer | Recruiting | Bladder cancer | 2 | NCT05531123 |
| 47 | Drug: neoadjuvant PD-1 inhibitor plus COX inhibitor; drug: neoadjuvant PD-1 inhibitor | Not described | | Toripalimab With or Without Celecoxib as Neoadjuvant Therapy in Resectable dMMR/MSI-H Colorectal Cancer | Recruiting | Colorectal cancer, dMMR, and MSI-H | 1\|2 | NCT03926338 |

**Table 1.** *Cont.*

| | Interventions | Period | Specific Timing | Title | Status | Conditions | Phase | Trial Identifier |
|---|---|---|---|---|---|---|---|---|
| 48 | Drug: Durvalumab Drug: Tremelimumab Drug: Enfortumab Vedotin Procedure: Radical Cystectomy | Not described | | Treatment Combination of Durvalumab, Tremelimumab and Enfortumab Vedotin or Durvalumab and Enfortumab Vedotin in Patients With Muscle Invasive Bladder Cancer Ineligible to Cisplatin or Who Refuse Cisplatin | Recruiting | Muscle-invasive bladder cancer | 3 | NCT04960709 |
| 49 | Drug: Nivolumab | Not described | | Pre-operative Immunotherapy in Stage II-III Urothelial Cancer | Active, not recruiting | Urothelial carcinoma | 1 | NCT04871594 |
| 50 | Drug: camrelizumab; drug: albumin paclitaxel; drug: cisplatin | Not described | | A Study of Perioperative Camrelizumab Combined With Chemotherapy in Patients With Resectable ESCC | Recruiting | Esophageal cancer | 2 | NCT05182944 |
| 51 | Drug: Tislelizumab | Not described | | Neoadjuvant PD-1 Monoclonal Antibody in Locally Advanced Upper Tract Urothelial Carcinoma | Unknown status | Locally advanced upper urinary tract urothelial carcinoma | 2 | NCT04672330 |
| 52 | Drug: Carboplatin; drug: Ipilimumab; drug: Nivolumab; drug: Paclitaxel; procedure: positron emission tomography and radiation therapy | Not described | | Nivolumab and Ipilimumab in Treating Patients With Esophageal and Gastroesophageal Junction Adenocarcinoma Undergoing Surgery | Suspended | Esophageal/gastroesophageal junction adenocarcinoma AJCC v8 stage II-III-IVA | 2\|3 | NCT03604991 |
| 53 | Drug: PD-1 antibody combined with FOLFIRINOX regimen; drug: PD-1 antibody combined with SOX program | Not described | | Immune Checkpoint Inhibitor PD-1 Antibody Combined With Chemotherapy in the Perioperative Treatment of Locally Advanced Resectable Gastric or Gastroesophageal Junction Adenocarcinoma | Recruiting | Locally advanced gastric adenocarcinoma | 2 | NCT04908566 |
| 54 | Drug: sintilimab; drug: Trastuzumab; drug: S-1 plus oxaliplatin | Not described | | SOX Combined With Sintilimab and Trastuzumab Versus SOX Regimen in the Perioperative Treatment of HER2-positive Locally Advanced Gastric Adenocarcinoma | Not yet recruiting | HER2-positive gastric or gastroesophageal junction adenocarcinoma | 2 | NCT05218148 |

**Table 1.** *Cont.*

| | Interventions | Period | Specific Timing | Title | Status | Conditions | Phase | Trial Identifier |
|---|---|---|---|---|---|---|---|---|
| 55 | Drug: Serplulimab, Albumin paclitaxel, and carboplatin AUC = 5; procedure: esophagectomy | Not described | | Serplulimab Combined With Chemotherapy in Patients With Resectable Esophageal Squamous Cell Carcinoma | Recruiting | Esophageal squamous cell carcinoma | 2 | NCT05659251 |
| 56 | Drug: PD-1 antibody; drug: Capecitabine; drug: Oxaliplatin radiotherapy | Not described | | The Combination of Hypofractionated Radiotherapy and Immunotherapy in Locally Recurrent Rectal Cancer | Not yet recruiting | Recurrent rectal cancer | 2 | NCT05628038 |
| 57 | Drug: neoadjuvant PD-L1 inhibitor | Not described | | Envafolimab as Neoadjuvant Immuntherapy in Resectable Local Advanced dMMR/MSI-H Colorectal Cancer | Recruiting | Colorectal cancer, dMMR, and MSI-H | 2 | NCT05371197 |

Clinical studies according to search results of the keywords "cancer" and "perioperative immunotherapy" in the past five years (since 1 January 2018) on https://clinicaltrials.gov/ (accessed on 14 April 2023).

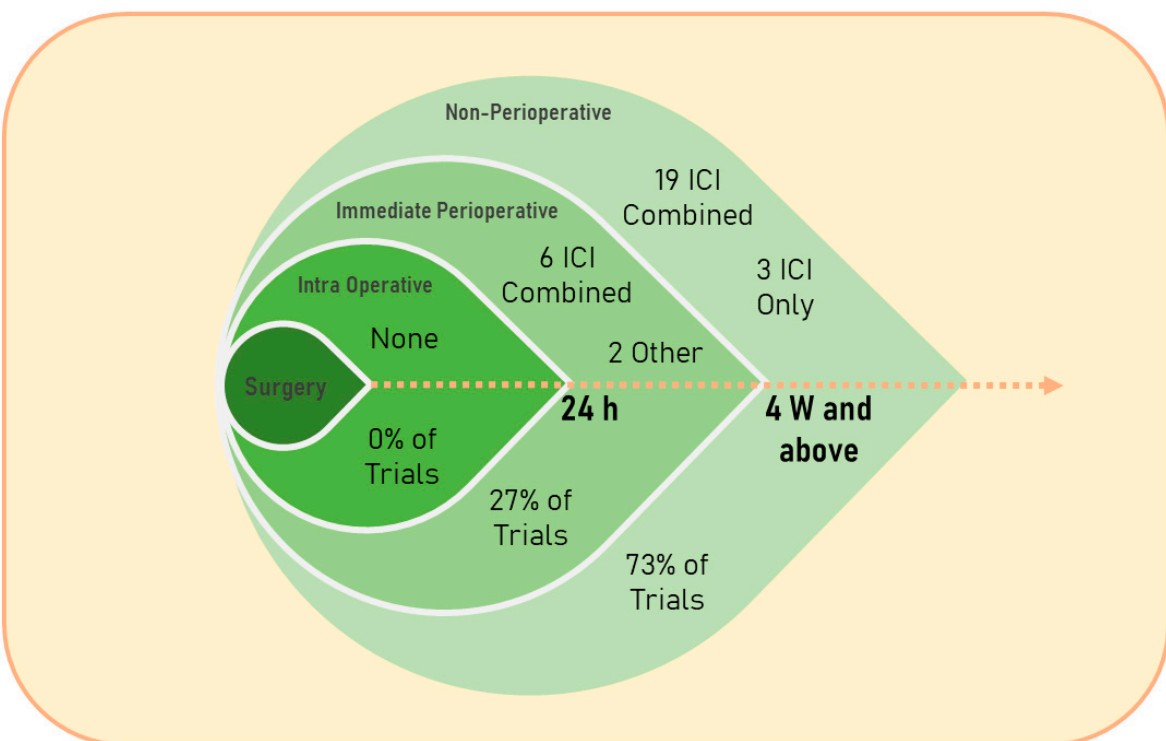

**Figure 1.** A schematic representation of Table 1. Timing relative to surgery and type of the treatment given to cancer patients in clinical trials found on search results of the keywords "cancer" and "perioperative immunotherapy" in the past five years (since 1 January 2018) on https://clinicaltrials. gov/ (accessed on 14 April 2023). 24 h, 24 hours; 4 W and above, 4 weeks and above; ICI, immune checkpoint inhibition; Combined, ICI therapy combined with other types of interventions.

Consequently, we employed this combined approach in the clinical setting, conducting two clinical trials in breast [11,69] and in colorectal cancer patients [70,71]. Propranolol and etodolac or placebo control were administered to the patients for up to twenty consecutive days, starting five days prior to tumor resection. Notably, such perioperative use of the drug combination exhibited a promising safety profile, with no adverse events and complication rates relative to the placebo-treated patients [11,69], including anticipated and manageable treatment-related adverse events, such as reduced blood pressure or heart rate [11,70]. Additionally, transcriptomic analysis of the excised breast and colorectal tumors revealed that compared to the placebo control group, propranolol and etodolac (i) reduced markers for epithelial-to-mesenchymal transition; (ii) decreased the activity of several transcription factors which are associated with inflammation and/or poor prognosis, including GATA1, GATA2, EGR3, STAT3, IRF1, IFR2, and CREB; and (iii) improved the composition of tumor-infiltrating leukocytes, including reduced monocytes and increased NK and B cell presence [11,70]. Importantly, the combined treatment reduced the plasma levels of prometastatic and proinflammatory proteins, including the plasma levels of IL-6 and CRP [11,70], even before surgery. Most importantly, we have now completed a 5-year follow-up in the colorectal cancer cohort, and reported a statistically significant improvement in disease-free survival, in both intent-to-treat and protocol-compliant patients [71]. However, this randomized clinical trial was planned as a biomarker study and did not have the statistical power to assess long-term cancer outcomes. Thus, we are now conducting larger and multicenter studies to further test the clinical long-term effectiveness of this novel approach, to advance our understanding of mediating mechanisms of the treatment, and to possibly introduce this prophylactic antimetastatic treatment as a clinical routine, alone or in conjunction with other therapy types.

### 5.1.2. Oncolytic Virotherapy

Oncolytic virotherapy (OVT) is a form of immunotherapy that employs a wide range of tumor-selective viruses, each of which target cancerous tissues [72]. OVT is designed to fight cancer by direct oncolysis and indirectly by stimulating a potent antitumor immune response [73]. Although still under investigation, OVT was shown to enhance antitumor immunity through the release of (i) tumor-associated antigens, (ii) pathogen-associated molecular patterns, and danger-associated molecular patterns, as well as (iii) cytokines (e.g., IFNγ, IL-12, and type-I IFNs) [74]. These effects enable oncolytic viruses to eliminate local residual tumor cells and distant micrometastases, as well as to impede angiogenesis, qualifying OVT as a neoadjuvant therapy [75,76].

According to Thomas and Bartee, there has been a recent surge in trials using neoadjuvant virotherapy, with four trials utilizing OVTs up to a month before surgery, and two trials ending treatment days before surgery, during the IPP [77]. Although factors determining the time of treatment proximity to surgery in these trials are not stated, a rationale for both distal and proximal intervals can be made. Evidence indicates that antitumor immunity is maintained long after the virus is cleared, allowing for extended intervals that can mitigate potential surgery-related risks. On the other hand, one study showed that OVT given three days before surgery led to a robust immune activation, potentially counteracting surgery-related immune suppression without evident complications [77]. T-VEC (Imlygic®, Amgen, CA, USA), a genetically modified herpes simplex virus, is the first and only FDA-approved OVT. It was recently studied in an open-label phase-II trial as a neoadjuvant therapy in stage IIIB–IVM1a melanoma [78]. Patients were assigned to (i) immediate surgery or (ii) intratumoral T-VEC treatment ending 1–4 weeks before surgery. The T-VEC arm demonstrated a significant 25% reduction in the risk of disease recurrence, with improved 2-year and 3-year relapse-free survival rates compared to the surgery-only group (29.5% vs. 16.5% and 28.1% vs. 16.9%) [78]. Another study has demonstrated how a single i.v. infusion of reovirus in a small cohort of patients with recurrent high-grade gliomas, 3–17 days before surgery, increased cytotoxic T cell tumor infiltration, upregulated IFN gene expression, and increased the PD-1/PD-L1 axis in tumors via an IFN-mediated mechanism, all of which were shown to potentially enhance the sensitivity of glioma cells to ICI treatment. Efficacy data, however, could not be determined due to the limited cohort size [79].

To summarize, OVTs demonstrate a generally well-tolerated safety profile, with the majority of adverse events being of low grade, and fewer high-grade adverse events compared to chemotherapy [80]. As OVT is still emerging and being developed, a qualitative comparison with other treatment modalities is not available, nor is the assessment of the significance of time interval to surgery. However, neoadjuvant OVT in the IPP continues to show encouraging results, particularly when used in combination with other immunotherapies [77].

### 5.1.3. Cytokine Therapies

Cytokines are defined as small proteins that are secreted from cells and are pivotal to cellular communication and interactions. Among other functions, these proteins are involved in controlling inflammation and are key components of an effective antitumor immune response [81]. Several cytokines have been studied clinically during the IPP as monotherapy or in conjunction with other therapies.

IFNα was the first cytokine-based therapy approved by the FDA in 1986 for patients with hairy-cell leukemia. Clinical studies evaluating IFNα administration during the IPP have found mixed results. For example, a combined treatment of IFNα and continuous arterial infusion of 5FU chemotherapy 2–3 weeks postoperatively increased the 1-year overall survival (OS) of hepatocellular carcinoma patients from 30% in patients who underwent surgery alone to 100% in patients who received the combined treatment [82]. Additionally, IFNα administration during the IPP reduced the postoperative suppression of NK cell cytotoxicity [83] and decreased the circulating levels of VEGF and the number

of regulatory T cells, all of which are related to a better prognosis [84]. However, a single dose of IFNα administered immediately following the transurethral resection of superficial bladder cancer did not improve OS compared to surgery alone [85].

Another promising cytokine in preclinical research, which has advanced to the clinical setting, is IL-2. IL-2 is a powerful regulator of immune responses [86] and is known to control the behavior of various leukocytes, especially T cells, promoting either antimetastatic or prometastatic effects in a dose-dependent manner [87]. Treatment with recombinant IL-2 has been studied since the 1990s, but has only been implemented during the IPP in very few studies. Nevertheless, these studies revealed encouraging results. For example, IL-2 therapy given twice daily for three consecutive days starting 5 days prior to surgery was shown to improve 5-year disease-free survival and OS in patients with pancreatic cancer [59] and reduced the frequency of disease progression in patients with colorectal cancer [60], compared with placebo. Moreover, IL-2 administered twice daily for five days until the day before surgery has demonstrated similar results [88]. Notably, the PANCEP-1 trial, evaluating peri- and postoperative treatment with histamine dihydrochloride (HDC) and low-dose IL-2 in patients with primary resectable pancreatic cancer, has recently been initiated. In this trial, patients receive HDC and IL-2 twice daily for three weeks, starting two weeks preoperatively up until the surgical resection, and continuing with the remaining week at three days postoperatively. The primary hypothesis for this trial is that the combined HDC and IL-2 treatment will reduce inflammation, counter surgery-related immunosuppression, and consequently reduce metastatic disease (NCT05810792).

Last, a combined treatment of nivolumab (anti-PD-1) seven days prior to surgery and VDX (a controlled IL-12 gene therapy) three hours before and fourteen days after surgery in patients with resectable recurrent glioblastoma has shown a great increase in serum IL-12 and tumor IFN-gamma postoperatively, leading to an ongoing phase-2 trial [62]. Nivolumab in conjunction with VDX treatment given during the IPP was manageable with regard to immune-related adverse events (irAEs), as they were reported to be dose-dependent, easily predicted, and reversible [62].

The safety profiles of these strategies were shown to be tolerable, with almost no surgery delays in neoadjuvant approaches and no increase in short-term or long-term surgical complications [60,88,89].

### 5.1.4. Monoclonal Antibodies

A common immunotherapeutic approach employs monoclonal antibodies (mAbs), which specifically bind to targets on tumor cells or stroma cells in the tumor microenvironment. Their binding can induce tumor lysis through different mechanisms, including a ligand or receptor blockade and the activation of host immunity [90,91]. Such mAbs can also be conjugated to a cytotoxic compound, such as chemotherapeutic agents, and some mAbs can simultaneously bind more than one target (known as bispecific antibodies) [90,91]. Common mAbs therapies in solid malignancies are usually given in cycles that can last weeks or even months. Although recent clinical research highlights the potential beneficial effects of neoadjuvant (preoperative) treatments, for example with anti-HER2 or anti-EGFR antibodies in BC [92], these are rarely administered during the IPP, with the neoadjuvant treatment regimen ending 3 weeks or more before surgery, and adjuvant therapy starting months after it [93–95]. Avoidance of mAbs administration during the IPP is attributed to potential adverse effects that the systemic delivery of mAbs may cause, even though mAbs immunotherapy is considered less toxic than traditional chemotherapy [96]. Immunotherapy with mAbs is associated with several toxicities, including hepatic, cardiac, pulmonary, renal, neurotoxicity, cytokine release syndrome, and infusion reactions [90,96,97].

Despite limited clinical research, preclinical studies show promise in the systemic delivery of mAbs during the IPP. Preclinical studies in models of BC have found that neoadjuvant anti-CD25 mAbs therapy targeting regulatory T cells given 3 days before surgery was significantly more beneficial in extending survival outcomes than if given 3 days after surgery [98]. The same group also found that ICI delivery was beneficial

in prolonging survival outcomes when given for short duration starting 4–5 days before surgery, but not if the treatment was terminated more than 8 days before surgery [99]. Extending therapy duration after surgery did not increase survival benefit, whereas it did induce more treatment-related adverse events [99]. Such preclinical studies pinpoint the importance of treatment duration and its timing relative to surgery. Additionally, in the setting of anti-HER2 mAbs therapy in BC patients, although pilot clinical trials assessing the delivery of neoadjuvant anti-HER2 mAbs therapy up to 7 days before surgery found it to be safe nearly 20 years ago [100], not many clinical trials employing such immunotherapy during the IPP have been conducted. A recent phase-II clinical trial evaluated targeted anti-HER2 therapy given at 11 and 3 days before surgery and approximately 2 weeks after surgery, and found that combining two drugs (lapatinib, a tyrosine kinase inhibitor, and trastuzumab, an anti-HER2 mAb) was more beneficial in reducing the proliferation marker Ki67 in the primary tumor within each patient (relative to presurgical biopsy samples), and also improved overall survival [61]. Of note, recent strategies are being developed for the intratumoral delivery of mAbs, in the form of mAbs gene delivery systems, through viral or nonviral vectors. Such strategies are expected to reduce unwanted adverse events which are observed during the systemic delivery of mAbs, as well as to affect distal metastases (either directly or indirectly), and these are currently being evaluated in both preclinical and clinical trials [101].

### 5.1.5. Immune Checkpoint Inhibitors: Treatment Evolvement and Usage within the IPP

ICIs have revolutionized cancer treatment, improving long-term cancer outcomes [102]. While ICIs are primarily administered in advanced and/or metastatic cancers, their perioperative use has recently gained considerable interest. Numerous studies have investigated the efficacy and safety of non-IPP use of ICIs in different cancer types, but almost none have evaluated ICIs during the IPP due to their confirmed and hypothesized adverse effects on the success of surgery and patients' safety [5]. Such adverse effects are commonly characterized and graded for severity by employing the Common Terminology Criteria for Adverse Events (CTCAE) introduced by the US National Cancer Institute [103]. The CTCAE grades irAEs on an ascending symptom severity scale: grade 1—asymptomatic/mild, grade 2—moderate, grade 3—severe, grade 4—life-threatening, and grade 5—death [103]. A systemic review and meta-analysis that assessed the general safety of ICIs in 36 phase-II and phase-III clinical trials estimated the pooled incidence rate of irAEs of any grade (1–5) to be between 66.4% and 86.8%, out of which grade-3–4 irAEs accounted for merely 14.1% and 28.6% (depending on the ICI type) [104], and grade 5 accounted for approximately 1% or less [105]. Although the incidence rate of ICI-therapy-related irAEs is relatively high, most studies employing these agents as monotherapy, dual therapy, or in conjunction with conventional therapy, describe a well-tolerated or manageable safety profile, especially when ICIs are administered for short durations in a limited number of cycles [106,107].

Treatment using ICIs was designed based on several key clinical trials in advanced and/or metastatic cancer patients, in multiple cycles and over prolonged treatment periods [107–109]. This strategy was adopted as the prevalent dogma, with the aim of eradicating MRD to prevent or postpone the development of metastatic disease [107], achieving FDA approvals for numerous indications [107]. While adjuvant therapy has been shown to mend pathologic responses in a significant portion of patients, some patients do not respond to treatment and some patients require an early cessation of treatment due to toxicities or disease hyperprogression [106]. Importantly, some studies assessed the long-term outcomes of patients who achieved pathological complete response (pCR) and ended the treatment early due to the manifestation of irAEs [110] or due to achieving pCR [111]. Unexpectedly, some patients who experienced a shorter treatment course continued to exhibit a durable response and had similar long-term benefits compared to patients who completed the entire treatment course, consequently raising questions about the necessity of a prolonged treatment for all patients, which is known to induce more irAEs [106,111].

Thus, further studies are warranted to assess the pathological responses and long-term benefits of shorter treatment courses of ICIs.

Studies that evaluate ICIs' efficacy in patients with early-stage cancer, which often involves surgical procedures in temporal proximity to the given therapy, have only gained momentum in recent years. A similar trend in other types of antimalignant therapy (such as chemo- and radiotherapy) has also occurred. The limited efficacy of adjuvant chemo- and radiotherapies in a significant proportion of cancer patients has led to the development of preoperative (neoadjuvant) treatment strategies. While adjuvant therapy commonly allows for longer treatment periods and enables a shorter diagnosis-to-surgery interval, neoadjuvant approaches have demonstrated significant advantages in several cancer types [112–115], including the following: (i) an increase in complete resection rates by the reduction in tumor load, (ii) the real-time evaluation of patient response to the treatment, before definitive surgery; and (iii) an increase in neoantigen presentation prior to surgery [116]. Additionally, there are indications that postoperative immunosuppression may hinder the efficacy of adjuvant immunotherapies. Accordingly, neoadjuvant approaches of ICIs have now been approved and are currently being studied in over 400 clinical studies (according to https://clinicaltrials.gov/ (accessed on 2 April 2023)). In these clinical studies, neoadjuvant ICIs are administered for a short period (up to 12 weeks) in a limited number of cycles. Encouraging results have already been demonstrated in several key studies. Notably, the CheckMate 816 trial evaluated the non-IPP neoadjuvant use of nivolumab and chemotherapy (ending four to six weeks before surgery), compared to chemotherapy alone, in resectable non-small-cell lung cancer (NSCLC) patients. The study demonstrated a significant improvement in median event-free survival (EFS) and a significantly higher pCR in the nivolumab and chemotherapy group (an EFS of 31.6 months vs. 20.8 months and a pCR of 24% vs. 2.2%). The treatment also maintained a satisfactory safety profile, as patients of both groups exhibited similar rates of adverse events and treatment-related adverse events [117]. Similarly, the NEOSTAR trial investigated the neoadjuvant Ipilimumab (anti-CTLA-4) vs. neoadjuvant nivolumab + ipilimumab in operable NSCLC patients, primarily assessing major pathological response (MPR). Although survival endpoints were not met, the results showed a significant increase in MPR, pCR, and in tumor-infiltrating nonregulatory T cells in the nivolumab + ipilimumab group. Toxicities were described as manageable, and grade >3 irAEs were reported in 13% of patients in the nivolumab + ipilimumab group vs. 10% in the Ipilimumab group [118]. These studies, like the great majority of clinical studies which evaluate the neoadjuvant use of ICIs, end the treatment before the IPP.

Preclinical studies assessing the timing of ICIs as adjuvant and neoadjuvant therapies have demonstrated the superior efficacy of neoadjuvant treatment in some cancer types, and especially when administered during the IPP. Liu et al. conducted a seminal study (which was briefly discussed above) in which they evaluated the timing relative to tumor resection and the duration of therapy of different ICI combinations in two models of murine breast cancer. Their results indicated that two doses of neoadjuvant anti-PD-1 + anti-CD137 therapy, starting 4–5 days prior to tumor resection, were advantageous over earlier neoadjuvant (10 days) or adjuvant therapies in expanding tumor-specific CD8+ T cells and prolonging survival rates. Nevertheless, when the treatment was initiated two days before surgery, the improvement in long-term survival was jeopardized, suggesting an optimal preoperative window of opportunity during the IPP. Interestingly, adding an adjuvant therapy component to the neoadjuvant therapy did not improve survival rates, but increased the irAE incidence rate [98,99], which again testifies to the importance of determining not only the treatment timing relative to surgery but also the treatment duration.

We were able to locate only two publications of completed clinical studies that evaluated neoadjuvant ICI therapy during the IPP, and both are small, single-arm, phase-I studies that assessed biomarkers and/or safety profiles. Nevertheless, these studies show a satisfactory safety profile and encouraging findings at the biomarker level. Specifically, Chiocca et al. used a combined treatment of nivolumab administered seven days prior to

surgery, and every two weeks following it, and VDX (controlled IL-12 gene therapy) given three hours before and fourteen days after surgery, in resectable recurrent glioblastoma patients. In this study, which was also reviewed briefly above, 21 patients were enrolled in three dose escalating cohorts to establish safe doses and biomarker response to the combined treatment. More than 60% of the patients exhibited grade-2 cytokine release syndrome, and nine patients exhibited nivolumab-related grade-3-and-above irAEs. Overall, treatment-related toxicities were considered tolerable and manageable as they were dose-dependent and easily predicted and reversed. This manageable safety profile and a postoperative increase in serum IL-12 and tumor IFN gamma levels led to an ongoing phase-II trial [62].

In the second trial, Huang and colleagues studied a single dose of neoadjuvant pembrolizumab (anti-PD-1), within three weeks preceding surgery, in stage IIIB/C or IV melanoma. Twenty-nine patients were enrolled to evaluate whether tumor immune reinvigoration would be evident within three weeks post-treatment (at surgery), and whether this response was correlated with long-term cancer outcomes. All patients were prescribed adjuvant therapy with pembrolizumab. The study met the a priori safety requirement of less than 30% grade >3 toxicities; there were no unexpected treatment-related irEAs, no unexpected delays in surgery or adjuvant therapy, and no surgical complications. Additionally, 27/29 tumor tissues were available for analysis, 8/27 achieved major or complete pathological response and are disease-free to this day, and the median 1-year survival rate of all patients was 63%. Furthermore, an analysis of tumor-specific T cell subsets in the blood at 7 and 21 days post-treatment, and in the tumor following excision (within 21 days post-treatment), indicated an expansion of circulating tumor-specific T cell subsets as early as 7 days post-treatment, and in the excised tumor tissues. These findings demonstrate the quick immunological response following anti-PD-1 therapy, which was also associated with improved long-term cancer outcomes within each subject [119].

Overall, ICI therapy in cancer patients is progressing rapidly; novel avenues of research are being explored, including an initial focus on the IPP, leading to an improvement in patients' responses to therapy and a consequent improvement in long-term outcomes. Few clinical trials are currently exploring ICI therapy during the IPP (see Tables 1 and 2), which constitute a very small portion of ICI research, and clearly, more trials are warranted to elucidate the feasibility and efficacy of ICI therapy during the IPP.

**Table 2.** Summary of clinical studies employing immune checkpoint inhibitors.

| Phase | Cancer Type | Patient N | Treatment | Timing | Citation |
|-------|-------------|-----------|-----------|--------|----------|
| I | Glioblastoma | 21 | ICI + GT | IPP | [62] |
| III | Melanoma | 676 | ICI | Non-IPP | [108] |
| III | Melanoma | 418 | ICI + CT | Non-IPP | [109] |
| I/II | Melanoma, NSCLC | 1260 | ICI | Non-IPP | [110] |
| Ib | Metastatic melanoma | 655 | ICI | Non-IPP | [111] |
| III | Bladder | 317 | CT | Non-IPP | [114] |
| III | Esophageal | 368 | CT + RT | Non-IPP | [115] |
| III | NSCLC | 505 | ICI + CT | Non-IPP | [117] |
| II | NSCLC | 44 | ICI | Non-IPP | [118] |
| I | Melanoma | 27 | ICI | IPP | [119] |

Summary of clinical trials mentioned in the above section employing ICIs during the IPP. ICI, immune checkpoint inhibitor; GT, gene therapy; CT, chemotherapy; RT, radiotherapy; NSCLC, non-small-cell lung cancer; IPP, immediate perioperative period.

## 5.2. Immunotherapy during the IPP: Limitations

The application of immunotherapies during the IPP presents new challenges to perioperative physicians, surgeons, and oncologists. The use of immunomodulatory agents during this period may (i) increase the incidence of irAEs during the IPP, (ii) hinder the success of surgery (e.g., impaired wound healing), and (iii) lessen the efficacy of immunotherapy

due to stress- and/or surgery-induced immunosuppression. Some of these challenges may necessitate the use of immunosuppressive drugs, may delay surgery, and potentially involve life-threatening complications [5].

For example, the use of immunomodulatory agents during the IPP, in conjunction with undetected surgery-related infection, may lead to an excessive inflammatory response, potentially resulting in SIRS (systemic inflammatory response syndrome) and organ failure [120,121]. Additionally, high-dose IL-2 treatment, but not short-course low doses [59,60], may induce a variety of adverse effects, including vascular leak syndrome, hypertension, renal failure, myocarditis, and thrombocytopenia, some of which may warrant a postponement of surgery and other supportive treatment [122,123]. Furthermore, treatment with IL-2 may induce an expansion of immunosuppressive regulatory T cell populations, which may hinder antitumor responses [122,123]. Similarly, ICIs may induce a range of systemic effects, depending on drug type, which can involve many organ systems, most commonly the skin, gastrointestinal, and endocrine gland systems, while uncommon and severe effects also involve pulmonary, cardiac, and neurologic systems [124,125]. The predominant method to treat the emergences of therapy-induced severe irAEs is the cessation of therapy and administration of immunosuppressants (e.g., glucocorticoids), both of which may hinder the efficacy of the anticancer therapy.

While most irAEs induced by the immunotherapies discussed in this paper were deemed tolerable and manageable [106,107], the constraints mentioned above, among others, warrant the development of standardized guidelines for the IPP use of immunotherapeutic agents. For the time being, intricate considerations of timing, duration, dosage, and drug combination, in addition to the utilization of existing toxicity management guidelines (e.g., SITC ICI toxicity management [126]) and assessment of patient- and organ-specific risk factors, should be considered for a clinical evaluation of immunotherapy during the IPP.

## 6. Conclusions

Immunotherapy has advanced tremendously during the past two decades, with an improvement in administration strategies and the development of new therapeutic approaches. Nevertheless, to avoid contraindications to surgery, the IPP remains relatively unexplored for antimetastatic immune interventions in cancer patients. The existing, yet limited, preclinical and clinical data demonstrate the safety and beneficial effects of immunotherapy during the IPP, yet further research is warranted, especially regarding the translation of preclinical findings into the clinical setting.

To minimize the gap between preclinical findings and their translational impact to the clinical setting, several key concerns and experimental settings should be taken into consideration. First, according to a recent review, fewer than 25% of studies that employed preclinical cancer models, studied processes related to metastasis, and out of these studies, fewer than 6% included a resection of a primary tumor, as in the clinical setting [127]. Thus, an optimal design of a translational preclinical study should mimic the clinically relevant surgical setting in which immunotherapy is administered. Second, studies that utilize immune-deprived mice (to allow the transplantation of human cancer cells) should also be conducted in immune-competent mice, especially in the context of immunotherapy, possibly alongside humanized models [128,129]. Third, preclinical studies that assess the impact of immunotherapy during the IPP on long-term cancer outcomes should account for several aspects that may jeopardize therapy potency, including presurgical stress [130], ambient temperature [25], treatment duration and timing relative to surgery [99], and drug release platform (e.g., systemic vs. local and injection vs. suspended release) [131]. Hopefully, these considerations will promote bench-to-bedside transitions of preclinical studies.

Using immunotherapy during the IPP, especially as a short-course neoadjuvant immunostimulatory application, may (i) circumvent cancer-treatment-induced immunosuppression (following chemotherapy, surgery, and sometimes radiotherapy), (ii) promote a rapid and nondelayed immunological response toward MRD, and (iii) avoid the deleterious

effects of surgery on pre-existing metastases while exerting minimal toxicities. Additionally, current biomarker strategies to assess patients' response to different types of immunotherapies, including the tumor neoantigen landscape, mutational burden, PD-L1 expression, and tumor-infiltrating lymphocyte landscape [132–134] in tumor and/or liquid biopsies [135], may enable accurate patient selection and improve the efficacy and safety of immunotherapy during the IPP. While pCR is considered a keystone in assessing clinical response to immunotherapy, it was shown, to some extent, to be inaccurate in predicting long-term clinical outcomes [136,137], in part due to neglecting partial responses such as MPR [137]. Thus, we suggest that future clinical studies should utilize multiple indications of clinical responses to achieve an accurate prediction of long-term clinical benefits alongside the routine approaches (e.g., pCR) [132,138–141].

Importantly, perioperative factors which may hinder the efficacy of immunotherapies, such as stress and inflammatory responses [2] and hypothermia [40,41], should also be considered. As such factors may vary individually, a patient-personalized approach is warranted. For example, patients who exhibit effective immune responses may benefit more from perioperative immunotherapy, and patients who experience high perioperative stress and immunosuppressive responses may benefit more from immunotherapy when combined with stress management (e.g., inhibition of adrenergic signaling and prostaglandin synthesis), as we have previously shown in preclinical studies [68,130]. Thus, future clinical studies assessing immunotherapy during the IPP may adopt biomarker-driven patient selection in combination with assessments of individual immune responsiveness and susceptibility to stress, while avoiding surgery- and hospitalization-related adverse effects.

Moreover, we believe that immunotherapeutic strategies (e.g., anti-stress-inflammatory approaches, ICIs, cytokines, TLR agonists, or oncolytic viruses) should be administered for a short and limited duration, and end or start as temporally close as possible to the surgery, considering the treatment efficacy and patient's safety. While some immunotherapies may be administered up to a day prior to surgery (e.g., IL-2) [88] or on the surgery day itself (e.g., inhibition of adrenergic signaling and prostaglandin synthesis) [71], some were shown in preclinical studies (as elaborated above) to be most effective and exhibit minimal toxicities when administered at a specific time point during the IPP (e.g., anti-PD-1 + anti-CD137) [98]. Thus, future clinical studies assessing immunotherapy use during the IPP, and specifically ICIs, should address the optimal timing of therapy during this critical timeframe.

Therefore, we suggest a combined approach to be tested in patients with early-stage resectable cancers, in which immunostimulatory treatment, such as ICI therapy, will be used for a limited duration (e.g., two cycles of two weeks), ending one to three weeks preoperatively. Additionally, complementing the latter, the inhibition of adrenergic signaling and prostaglandin synthesis during the week before and after surgical resection—in order to attenuate stress and inflammation, respectively—may be used to mitigate perioperative immunosuppression and the direct stimulation of MRD. Potentially, combinations of these drugs, when given in a relative temporal proximity to the surgery and for a short duration, may improve the efficacy of the immunostimulatory agent and may promote high response rates in patients while exerting minimal toxicities. The utilization of such therapy combinations during the IPP may constitute a novel treatment regimen to improve the long-term outcomes of cancer patients.

**Author Contributions:** Conceptualization, E.S. and S.B.-E.; writing—original draft preparation, E.S., S.B.-E.; writing—review and editing, E.S., A.E., A.M., L.S. and S.B.-E.; funding acquisition, S.B.-E. All authors have read and agreed to the published version of the manuscript.

**Funding:** This research was funded by the "Israel Cancer Research Fund" (ICRF), grant number PG-20-106, and the "Israel Ministry of Health" (IMH), grant number 3-17323.

**Institutional Review Board Statement:** Not applicable.

**Informed Consent Statement:** Not applicable.

**Data Availability Statement:** Not applicable.

**Conflicts of Interest:** The authors declare no conflict of interest.

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
