# Peer review of "Immunotherapy during the Immediate Perioperative Period: A Promising Approach against Metastatic Disease"

_curroncol, doi:10.3390/curroncol30080540_

Round 1

Reviewer 1 Report

The manuscript is well written, results are clear, and the topic is highly interesting. In this review authors discuss safety and potential efficacy of different immunotherapies in the perioperative context, and which types of available immunotherapies are most suitable. The rationale of this review assumes that operated cancer patients are subjected to a stress and inflammatory responses throughout their disease course, from the diagnosis, through numerous diagnostic and surgical procedures. The biological responses to surgery and its related psychological stress were shown to promote numerous signaling cascades, including sympathetic, steroid, inflammatory, and pro-angiogenic responses, which in-turn could lead to immune suppression. Notably, β-adrenergic and prostaglandin signaling were shown to promote tumor cells epithelial-to-mesenchymal transition by inducing secretion of pro-angiogenic factors (VEGF, IL- 87 6, IL-8) in human melanoma, breast, and ovarian cancer cells. Immunotherapy achieves its goals by: employing cytokines or Toll-like receptor agonists; preserving anti-cancer immune functions through blockade of immunosuppressive mediators, or through inhibition of immune-checkpoint signaling, or programmed cell death 1 (PD-1) inhibition; directing oncolytic viruses or monoclonal antibodies at the malignant tissue to induce immunogenic cell death, or to enhance immune responses to tumor cells; For example, recent studies showed how the strategy of the VEGFR-2-targeted therapy should be considered in the treatment of the VEGF/VEGFR-2-associated diseases by blocking the VEGF/VEGFR-2 signaling pathway, inhibiting VEGF and VEGFR-2 gene expression, blocking the binding of VEGF and VEGFR-2, and preventing the proliferation, migration, and survival of vascular endothelial cells expressing VEGFR-2.

-       Please mention the following study in the discussion section: Wang X, Bove AM, Simone G, Ma B. Molecular Bases of VEGFR-2-Mediated Physiological Function and Pathological Role. Front Cell Dev Biol. 2020 Nov 16;8:599281

In preclinical studies a short-course neoadjuvant immunotherapy may circumvent treatment-induced immunosuppression, promote a rapid and non-delayed immunological response directed at minimal residual disease and avoid the deleterious effects of surgery on ongoing metastasis. Despite the promising results of pre or post-operative immunotherapy treatments, further studies are needed to assess the clinical advantage of such treatment in an early post-operative setting as a standard pathway.

Reviewer 2 Report

I would like to thank the handling editor for offering me the opportunity to review the manuscript entitled “Immunotherapy during the immediate perioperative period: A promising approach against metastatic disease” authored by Sandbank and colleagues, which is currently under consideration for publication in the Current Oncology. I would also like to commend the authors for their scholarly work, which presents a narrative review examining how immunotherapy could be leveraged to improve cancer outcomes during the critical immediate perioperative period surrounding tumour resection surgery. The immediate perioperative period, defined as 3 weeks prior to and following surgery, represents a pivotal window that impacts long-term prognosis. Surgical trauma and related psychological stress promote immunosuppression and progression of minimal residual disease. However, immunotherapy is rarely implemented during the IPP window due to concerns regarding safety and potential contraindications to surgery. Overall, this review makes the case that the under-explored immediate perioperative period timeframe represents an opportune window for immunotherapy that warrants further investigation.

The manuscript under review has the potential to make a valuable contribution to the existing literature on perioperative care and immunotherapy for cancer patients. The authors comprehensively review the critical, yet under-explored, timeframe of the immediate perioperative period as an opportune window for immunotherapy. Their thesis that carefully timed immunotherapy during this brief period may improve long-term outcomes is relatively novel and impactful.

The manuscript is strengthened by its thorough review of current evidence. The authors synthesize preclinical and emerging clinical data demonstrating the safety and efficacy of select immunotherapies when administered perioperatively. They make a compelling case that the immediate perioperative period represents an immunological inflection point, where surgery-induced immunosuppression and minimal residual disease progression might be mitigated by properly timed immunomodulation.

The authors thoughtfully consider the feasibility and promise of various therapeutic approaches, including anti-stress agents, oncolytic viruses, cytokines, monoclonal antibodies, and immune checkpoint inhibitors. The review is expertly situated within the current resurgence of immunotherapy across multiple cancer indications. Discussion of recent neoadjuvant trials provides important clinical context.

By evaluating the potential of the immediate perioperative period as an adjuvant window for immunotherapy, this manuscript meaningfully advances current thinking at the intersection of surgical, immunological, and psychological considerations in oncology. The authors reinforce the urgent need for further clinical investigation through well-reasoned arguments and a forward-looking perspective. Their novel framework serves as a launching point for research that could ultimately improve patient outcomes.

While the manuscript provides valuable insights, there are a few areas that could be refined to further augment the quality and impact of the work. Here are some respectful suggestions that could potentially enhance the manuscript:

1.      A separate and concise methods section would strengthen this narrative review. For instance, the authors may consider reporting the databases searched and date ranges covered, key search terms, and criteria for study selection. Outlining even basic methods would provide context on the rigorousness and reproducibility of the literature search and selection process undertaken for the review.

2.      To aid the transition to human studies, the authors may consider a section reviewing key preclinical models and knowledge gaps that future animal research could target. This could bridge the bench-to-bedside divide.

3.      When discussing clinical data, the addition of summary tables could allow quicker evaluation of the existing evidence base across perioperative immunotherapy trials. This may be especially helpful for the immune checkpoint inhibitor section.

4.      Expanding the discussion of clinical trial design considerations for perioperative immunotherapy may further strengthen the work. The authors could elaborate on optimal timing, inclusion criteria, and endpoints for future studies. Proposing specific next steps would aid translation of this novel framework into impactful trials.

5.      More comparison of the reviewed therapeutic approaches in terms of mechanisms, immune effects, and strength of clinical evidence may help readers evaluate the relative promise of each modality. Tables could visually summarize these factors.

6.      The authors could comment on the potential for biomarker-driven patient selection and response prediction regarding perioperative immunotherapy. This advancing area may have implications for clinical trial enrichment and personalized oncology.

7.      The authors may consider adding a brief limitations section to promote a balanced perspective. This could acknowledge inherent constraints in methodology and comprehensiveness for any review format, as well as uncertainties regarding the evidence included in the present review.

In conclusion, I would like to reiterate my appreciation to both the editor and the authors for the opportunity to review this intriguing and informative manuscript. I trust that my suggestions will help enhance the robustness and relevance of this important work. I look forward to seeing the revised version of the manuscript and wish the authors success in their ongoing research endeavours.

Overall, the manuscript exhibits high quality English language and academic writing style. Some minor areas for improvement may include the following:

  • More judicious use of abbreviations could aid readability for a multidisciplinary audience.
  • Paragraph length could be more consistent in certain sections.

Round 2

Reviewer 2 Report

I want to express my appreciation for the attention and consideration you have devoted to my suggested revisions for your manuscript. It is evident that a significant amount of effort and thought has been directed towards the refining of your work, integrating the feedback provided during the peer review process. The resulting modifications demonstrate a thorough and thoughtful approach, and significantly enhance the rigor and overall quality of your manuscript. I look forward to witnessing the impact your research will undoubtedly have on the academic community.

The manuscript demonstrates exemplary English language usage and academic writing style, making it well-suited for publication in this prestigious journal.